# Multiple Bioactivities of *Manihot esculenta* Leaves: UV Filter, Anti-Oxidation, Anti-Melanogenesis, Collagen Synthesis Enhancement, and Anti-Adipogenesis

**DOI:** 10.3390/molecules27051556

**Published:** 2022-02-25

**Authors:** Manuschanok Jampa, Khaetthareeya Sutthanut, Natthida Weerapreeyakul, Wipawee Tukummee, Jintanaporn Wattanathorn, Suparporn Muchimapura

**Affiliations:** 1Department of Pharmaceutical Chemistry, Faculty of Pharmaceutical Sciences, Khon Kaen University, Khon Kaen 40002, Thailand; manuschanok_j@kkumail.com (M.J.); natthida@kku.ac.th (N.W.); 2Human High Performance & Health Promotion Research Institute (HHP&HP Research Institute), Khon Kaen University, Khon Kaen 40002, Thailand; wipath@kku.ac.th (W.T.); jintanapornw@yahoo.com (J.W.); supmuc@kku.ac.th (S.M.); 3Department of Physiology (Neuroscience Program), Faculty of Medicine, Khon Kaen University, Khon Kaen 40002, Thailand

**Keywords:** *Manihot esculenta*, cassava leaf, UV filter, flavonoids, melanin, collagen, adipogenesis

## Abstract

The cassava root is an important global agro-industrial crop that yields cassava leaf as a left-over co-product of interest for further development as a sustainable resource of health and cosmeceutical active compounds. This work aimed to investigate the cosmeceutical potential and chemical composition of an ethanolic cassava leaf extract (BM). rutin, apigenin, and kaempferol were found to be major constituents via HPLC-DAD UV analysis. Interestingly, the multiple beneficial bioactivities of BM for cosmeceutical applications were manifested in a dose-dependent manner, including anti-oxidation in a 2,2-diphenyl-1-picrylhydrazyl assay, anti-melanogenesis in B16 melanoma cells, collagen synthesis enhancement in human fibroblasts, and anti-adipogenesis in 3T3-L1 adipocytes. Furthermore, the potential of the collagen synthesis enhancement of BM and rutin was significant when compared to ascorbic acid. Additionally, a UV filter property comparable to BEMT with characteristics of board spectral absorption and constant high absorptivity throughout all UV wavelength ranges was exhibited by UV-visible spectrophotometric analysis. In conclusion, the cassava leaf was found to be a potential natural cosmeceutical active agent with multiple cosmeceutical-related bioactivities with respect to a substantial composition of bioactive flavonols. These obtained data will support and encourage the further study and development of cassava leaves as potential economic and sustainable sources of bioactive agents for health and cosmeceutical applications.

## 1. Introduction

*Manihot esculenta* (L.) Crantz in the Euphorbiaceae family (cassava) is an important global agro-industrial crop, whose roots are used for food product manufacturing. Meanwhile, cassava leaves are a left-over co-product in substantial amounts with limited utility and supporting scientific data. Although traditional Nigerian remedies have used cassava leaves to relieve fever, headache, rheumatism, hemorrhoids, ringworm infection, conjunctivitis, sores, and abscesses [1], this has been practiced with a lack of supporting scientific data. Nevertheless, interestingly, the health benefits of cassava leaves have been suggested by reports on their nutritional, mineral, and phytochemicals contents: carbohydrates, essential amino acids (methionine and phenylalanine), dietary fibers, β-carotene, vitamins, phenolics, anthocyanins, and flavonoids [2,3]. Furthermore, the flavonol rutin was also proposed as a bioactive constituent contributing to its antioxidative property [4]. Therefore, to support the rational utility and maximize the economic value of cassava leaves, scientific data regarding their bioactivity, cosmeceutical potential, and health benefits are crucial.

The global aging society and consumers’ concerns about their physical appearance have driven the continuing growth of the cosmeceuticals market. Particularly, skincare products containing natural cosmeceuticals with multiple targeting effects, safety, and sustainability for industrial production have emphasized the increasing demand and market share in the cosmetic and personal care industry. Skin aging is an aesthetic problem caused by intrinsic- and extrinsic-factor-induced cellular adulteration—hyper-melanogenesis in melanocytes and a reduction in collagen synthesis in fibroblasts—leading to black spots, wrinkles, and skin sagging. Solar ultraviolet (UV) rays or sources of UV radiation, particularly the most abundant radiation, UVB (290–320 nm), and the powerful skin-penetrating UVA (320–400 nm) region, are significant influencing factors of premature skin aging or photoaging. They can dramatically accelerate the skin aging process [5] through UV-induced reactive oxygen species (ROS) overproduction. Subsequently, this can lead to the stimulation of the melanogenesis pathway in melanocytes in the *Stratum basale* layer and collagen degradation, but also the attenuation of collagen synthesis, resulting in skin dehydration, loss of flexibility, and wrinkles [6]. Thus, UV filters, anti-oxidation, anti-melanogenesis, and collagen synthesis enhancement are the empirical cause-solving mechanisms of cosmeceutical agents [7]. In addition to skin-aging problems, oxidative-stress-related obesity and cellulite development have become further intricate aesthetic concerns among consumers. Therefore, prevention and treatment through anti-adipogenic effects that inhibit the preadipocyte differentiation and lipid accumulation of anti-obesity agents have been proposed [8,9,10,11]. 

As cause-solving elements, effective cosmeceutical agents with multi-bioactivity that can regulate multiple targets of action in dermal cells are currently applied in pursuit of holistic outcomes. In addition, holistic outcomes contribute to rational and complex phytochemical compositions in natural extracts that possess multiple bioactivities, and have drawn significant attention from consumers and manufacturers. For example, phenolics and flavonoids are acknowledged for their cosmeceutical activities and are used in numerous anti-aging products; these cover UV filtering, anti-oxidation, anti-inflammation, anti-melanogenesis, collagen degradation inhibition, and the stimulation of collagen synthesis [11,12,13], which are safe and sustainable for long-term use. Therefore, this study aimed to investigate the cosmeceutical potential of cassava leaves, focusing on their phytochemical composition and cosmeceutical effects to encompass further research and development, which ultimately lead to sustainable and rational utility.

## 2. Results

### 2.1. Chemical Composition by HPLC-UV DAD Analysis

The complex composition of ethanolic cassava leaf extract (BM) was demonstrated by the obtained HPLC chromatogram, which comprised multiple peaks at different retention times (RT), each with specific spectroscopic characters (as shown by the maximal absorption with a specific wavelength; λmax). Among them, three major peaks were found. Compared to the standard compounds and following the principle of compound identity on physicochemical and spectroscopic characters, rutin (RT 23.61 min, λmax 213/256/355 nm), kaempferol (RT 24.66 min, λmax 264/350/395 nm), and apigenin (RT 31.88 min, λmax 265/335/396 nm) were the major chemical constituents of BM (Figure 1A). Their contents were ranked as rutin (39.96 ± 0.81 mg/g extract), apigenin (27.70 ± 0.40 mg/g extract), and kaempferol (15.73 ± 0.19 mg/g extract) (Table 1). Additionally, other minor peaks of unidentified compounds (1–10) and Gallic acid (λmax 221/271 nm) at RT 12.13 min were also detected. Interestingly, a peak (Unknown 5) detected at RT 33.16 min (λmax 265/333/396 nm) exhibited a similar spectroscopic profile to that of the apigenin (RT 31.88 min, λmax 265/335/396 nm); however, it was detected at different RTs. By contrast, Unknown 9 and 10 displayed high absorption in the visible wavelength region (400–700 nm) (Figure 1B). 

### 2.2. UV Absorption Covering UVA and UVB Rays

The absorption spectra of a reference UV filter (BEMT) covered the UV wavelength region 200–400 nm with lambda max (λmax) at 316 and 367 nm—wavelengths with maximal absorption. Relatively, the absorptivity ($A_{1-cm}^{1\%}$) of the BEMT at each λmax wavelength was delivered from the corresponding linear equation: primary band at 316 nm with $A_{1-cm}^{1\%}$ = 503.53, and secondary band at 367 nm with $A_{1-cm}^{1\%}$ = 417.54 (Figure 2A1,A2). Interestingly, the UV spectral absorption of the ethanolic cassava leaf extract (BM) differed from that of the BEMT. The BM exhibited a broadband absorption covering both UV and visible regions (wavelength 200–700 nm), in which the λmax(s) extended in the UVA (320–400 nm) and UVB (290–320 nm) wavelength regions (Figure 2A1,B1) with consistent absorptivity (290–400 nm). There was a primary band at 269 nm with $A_{1-cm}^{1\%}$ = 33.136, a secondary band at 363 nm with $A_{1-cm}^{1\%}$ = 28.881, and a tertiary band at 355 nm $A_{1-cm}^{1\%}$ = 27.534 (Figure 2B1,B2). Moreover, the UV absorption capacity of BM at a concentration of 500 µg/mL was comparable to that of BEMT (at 25 µg/mL) (Figure 2C).

### 2.3. Anti-Oxidation

The anti-oxidation of the ethanolic cassava leaf extract (BM) was investigated in comparison to standard antioxidants: ascorbic acid—the representative of reducing antioxidants via electron transferring (reducing potential); BHT—a synthetic phenolic antioxidant via hydrogen atom transferring (HAT) to neutralize the DPPH• free radicals [14]. The moderate-degree antioxidative effect of BM was demonstrated in a dose-dependent manner with an IC_50_ value of 319.37 ± 2.60 µg/mL when compared to the standard antioxidants: ascorbic acid (IC_50_ 3.99 ± 0.13 µg/mL) and BHT (IC_50_ 688.16 ± 9.22 µg/mL) (Figure 3).

### 2.4. Anti-Melanogenesis in B16 Melanoma Cells

The bi-modular effect of the ethanolic cassava leaf extract (BM) on melanogenesis was detected. With concentration dependency, a slightly increased melanin content was found at a low concentration range of the BM treatment (lower than 62.5 μg/mL); by contrast, a significant reduction in melanin content was achieved at high concentrations of BM treatment (250–500 μg/mL) with melanin contents of 82.13 ± 2.95 and 67.22 ± 1.83% of the control, respectively. Additionally, the results showed the potent anti-melanogenesis effect of Kojic acid in a dose-dependent manner with significance at a concentration range between 2.5 and 20 μg/mL. Meanwhile, the BEMT, the reference UV filter, showed no activity at any of the tested concentrations, with melanin contents ranging between 97.03 ± 3.28 and 101.36 ± 3.24% of the control (Figure 4).

### 2.5. Collagen Synthesis Enhancement in NHDF Fibroblasts

The results showed the non-cytotoxic impact of all the samples at the tested concentrations (% viability > 80, compared to the control (untreated) group). In addition, the collagen synthesis enhancement of the ethanolic cassava leaf extract (BM), rutin, and vitamin C was significant when increasing the collagen content after treatment. The BM exhibited the highest potency of action, showing significant effects on cell proliferation (reflected by a higher cell viability—113% of the control) and collagen synthesis induction in fibroblast cells (collagen content as high as 281.87 ± 8% of the control), while rutin (233.52 ± 2.62% of the control) and vitamin C (186 ± 8.48% of the control) showed a strong collagen synthesis-enhancing effect, but not on fibroblast proliferation (cell viability of 90–100% of the control). By contrast, no effect of BEMT on fibroblast collagen synthesis was detected (Figure 5).

### 2.6. Anti-Adipogenesis in 3T3-L1 Adipocytes

The anti-adipogenesis of the ethanolic cassava leaf extract (BM) was demonstrated by the diminishing Oil Red O-stained oil droplet numbers and sizes after the BM treatment at various concentrations (Figure 6A). This suggested an inhibitory effect on adipocyte differentiation―a process of preadipocyte transformation into mature adipocytes resulted in lipid-droplet-filled and rounded-shape morphological characters prior to intracellularly synthesizing and cumulating the lipid content in the form of triglycerides. Relatively, the anti-adipogenic effect was confirmed by the reduction in the intracellular lipid content after BM treatment in a dose-dependent manner, with lipid accumulation in the range of 59.17 ± 0.94 to 99.64 ± 0.86% of the control (with % cell viability > 80) (Figure 6B). The effects were significant at BM concentrations of 500 and 1000 μg/mL, with lipid accumulation of 78.63 ± 0.45 and 59.17 ± 0.94% of the control. 

## 3. Discussion

The overall results of this research demonstrated the cosmeceutical potential of cassava leaves, with multiple bioactivities and a rational phytochemical composition of flavonoids as their major constituents. These data were considered as early supporting evidence of the beneficial bioactivities of cassava leaves for health and aesthetics. In addition, their inexpensive and highly left-over quantities as co-products of cassava root production support their promising industrial feasibility regarding economics and sustainability. The HPLC analysis demonstrated the complexity of chemical composition in the cassava leaf extract using an HPLC chromatogram composed of more than 13 peaks of compounds. Interestingly, three predominant peaks with typical UV absorption profiles of flavonoids were detected and partially identified as rutin (39.96 mg/g extract), apigenin (27.70 mg/g extract), and kaempferol (15.73 mg/g extract), which made the overall content as high as 83.39 mg/g extract (~8% *w*/*w*). These results agreed with a previous report on the dominant flavonoid content in a cassava leaf sample composed of rutin (58.89%; 6.22 mg/g) and nicotiflorin or kaempferol 3-O-β-rutinoside (29.31%; 3.09 mg/g), as well as some other flavonoids (0.96–4.81%; 0.10–0.50 mg/g), such as myricetin-3-O-rutinoside, robinin, hyperoside, nicotiflorin, and narcissin [3]. Additionally, other classes of phytochemical, such as phenolics and phytopigments, were also detected in cassava leaves. With respect to the large amount of evidence on the bioactivities of flavonoids, they have been suggested as potential bioactive markers of cassava leaves. However, only a few applicable supporting data have identified the chemical constituents and related bioactive markers of cassava leaves. Thus, future in-depth studies identifying the bioactive markers of cassava leaves should be encouraged to facilitate the standardization of the raw material and extracts for industrial manufacturing and applications. 

Sunscreen application containing UV filters is a practical prevention approach to skin photoaging [11,15,16,17]. However, UV-filter skin-penetration-induced allergies and dermatitis are concerning issues [18]. As potential solutions, natural extracts composed of an appropriate combination of phenolics, flavonoids, and phytopigments are effective alternative UV filters of interest, which provide a wide range of UV absorptivity and antioxidative activity to counteract UV-induced ROS generation and skin aging progression. BM was found as a natural UV filter of interest with constant high absorptivity throughout all UV wavelength ranges, including the UVC (200–280 nm), UVB (280–320 nm), and UVA regions (320–400 nm) (Figure 2). In addition, the constantly high absorptivity throughout the 250–400 nm UV wavelength region of BM implied an advantageous UV filter property (Figure 2A1), presumably associated with the spectroscopic property of major constituents such as rutin, apigenin, and kaempferol, which effectively absorb UV rays in the 260–380 nm wavelength region (Figure 1A). Additionally, the absorption could extend to the visible wavelength region (400–700 nm). This was partially supported by the reports of phytopigment composition in the cassava leaf [3,19]. Compared to the BEMT―a popular broadband UV filter in sunscreen products—it showed no cosmeceutical effects and varying absorptivity along the UVA and UVB wavelengths. These data emphasize the benefits of cassava leaf extract for cosmeceuticals: UV filtering and multiple biological effects. In addition to the UV absorption profile, the photo-stability and optimal concentration for an efficacious UV filter of cassava leaf extract should be further evaluated. However, the photo-stability of the three major constituents―rutin, apigenin, and kaempferol—has been addressed [20,21]. In addition, by forecasting the UV filter efficacy and optimizing the effective concentration of BM, the absorptivity coefficient ($A_{1-cm}^{1\%})$—the value that describes the absorptivity at a specific wavelength under conditions of 1% concentration and 1 cm light path width—is considered a beneficial parameter to facilitate retrieval based on the calculation. Moreover, $A_{1-cm}^{1\%}$ would be useful for the standardization and quantitative analysis of BM extract in industrial manufacturing. 

As a piece of early evidence on the cosmeceutical potential of cassava leaves, the multiple bioactivities of BM and its rational composition and content of the well-known bioactive flavonols (rutin, apigenin, and kaempferol) were demonstrated in our study results (Figure 1). The benefits of these flavonoids due to their wide spectral beneficial bioactivities for use in cosmetics or pharmaceuticals are widely acknowledged. The desirable UV filter and anti-oxidation of flavonoids fighting against UV-induced skin aging and damage caused by oxidative stress after chronic UV ray exposure [22,23] support our results, which demonstrated the superior antioxidative effect of BM compared to BHT—a chemical antioxidant with a HAT mechanism of action—in a DPPH assay. The synergistic effect between the chemical constituents in BM was proposed as the underlining factor of the obtained result. This hypothesis was partially supported by a report on the synergistic antioxidative effect of quercetin, gallic acid, and rutin, in which their mechanisms via both HAT and chelating were reported [16,24,25,26,27,28]. Furthermore, the multiple bioactivities of BM and flavonoid composition have potential for holistic cosmeceutical outcomes [29,30]. This is strongly supported by numerous previous reports on the cosmeceutical effects and biomolecular mechanisms of rutin, apigenin, and kaempferol. Rutin and apigenin regulated skin collagen synthesis and degradation through the attenuation of phosphoinositide 3-kinase (PI3K), resulting in the incretion of collagen type I (COL1A1) content and the suppression of metallopeptidase-1 (MMP-1) expression, which decreased the collagen degradation from a responsible enzyme, MMP-1 [16,17,27,31,32], as well as anti-melanogenesis, via tyrosinase inhibition [33,34,35,36]. Accordingly, our results illustrated similar manners of increasing synthesized collagen contents between BM and rutin with dose dependence and significant potency (Figure 5). Interestingly, a synergistic effect among the BM chemical constituents was suggested from the results demonstrating the greater effects of BM at a 250 μg/mL concentration (estimated rutin content of 1 μg) than rutin at a 50 μg/mL concentration. In addition, kaempferol and derivatives exhibited strong anti-oxidation and anti-melanogenesis effects, related inhibitory extracellular tyrosinase activities, and decreasing intracellular tyrosinase activities, as well as the downregulation of tyrosinase (TYR) and tyrosinase-related protein 1 (TRP-1) expression as the biomolecular targets [37,38]. This information supports the use of kaempferol and derivatives and related natural sources for cosmeceutical purposes. 

The potential of BM for anti-obesity and anti-cellulite applications has been suggested due to its anti-adipogenesis effect and the chemical composition of rutin, apigenin, and kaempferol, the anti-adipogenic effects and biomolecular mechanisms of which have been evidenced. Rutin can inhibit differentiation and adipogenesis via a reduction in triglyceride accumulation in preadipocytes and the suppression of such adipogenic-regulatory genes as peroxisome proliferator-activated receptor γ(P-PARγ), CCAAT/enhancer-binding protein (C/EBPα), and leptin1 [6,39,40]. Apigenin can regulate this phenomenon [41,42,43] with an extensive range of biomolecular targets, including the activation of 5’-adenosine monophosphate-activated protein kinase (AMPK), the inhibition of mitotic clonal expansion, G0/G1 cell cycle arrest, the suppression of the signal transducer, and the activation of the transcription 3 (STAT3) target gene cluster of differentiation 36 (CD36) [44]. Kaempferol and its derivatives possess an anti-adipogenesis effect via the anti-differentiation and inhibition of lipid accumulation. Similar to rutin, the anti-adipogenic effects of kaempferol and its derivatives, such as kaempferol-3-*O*-rutinoside and 3-*O*-[(*E*)-4-(4-cyanophenyl)-2-oxobut-3-en-1-yl] kaempferol, were shown. Kaempferol and kaempferol-3-*O*-rutinoside could reduce lipid accumulation in 3T3-L1 adipocytes by decreasing the gene expression of PPARγ, C/EBPα, and SREBP-1c [45,46]. The 3-*O*-[(*E*)-4-(4-cyanophenyl)-2-oxobut-3-en-1-yl] kaempferol was an in vivo anti-obesity and anti-diabetes agent for which biomolecular mechanisms were defined regarding the inhibition of intracellular lipid accumulation and the adipocyte differentiation of 3T3-L1 cells by the downregulation of peroxisome proliferator-activated receptor gamma (PPARγ) and CCAAT/enhancer binding protein α (C/EBPα) gene expression, the induction of miR-27a/b gene expression, and the activation of AMP-activated protein kinase (AMPK) activity [47]. 

Taken together, the overall results and previous evidence show the fascinating potential of cassava leaves as natural cosmeceutical agents due to the advantages of their multiple bioactivities, including UV filtering, anti-oxidation, anti-melanogenesis, collagen synthesis enhancement, and anti-adipogenesis. Furthermore, rutin, apigenin, and kaempferol are proposed as tentative bioactive markers for the standardization and quality control of cassava leaves and products. However, further studies delineating their efficacy, mechanisms, and safety in in vivo systems are encouraged to support the rational application, added value, and sustainability of cassava leaf usage.

## 4. Materials and Methods

### 4.1. Reagents and Instruments 

Ascorbic acid, gallic acid, rutin, kaempferol, apigenin, beta-sitosterol, stigmasterol, biotin, dimethyl sulfoxide (DMSO), methanol (HPLC grade), penicillin, streptomycin, butylated hydroxytoluene (BHT), thiobarbituric acid (TBA), dexamethasone (DEX), 2,2-diphenyl-1-picrylhydrazyl (DPPH), 3-isobutyl-1-methylxanthine (IBMX), butylated hydroxytoluene (BHT), insulin, Oil Red O solution, and forskolin were purchased from Sigma-Aldrich (St. Louis, MO, USA). Hexane, methanol, and ethanol (analytical grade) were purchased from RCI Labscan (Bangkok, Thailand). Bis-ethylhexyloxyphenol methoxyphenyl triazine (BEMT; bemotrizinol or Tinosorb S) was purchased from Chanjao Longevity Co., Ltd. (Bangkok, Thailand). The antibiotic-antimycotic solution containing penicillin, streptomycin, and amphotericin B, Dulbecco’s modified eagle medium (DMEM), fetal bovine serum (FBS), and 0.25% trypsin was purchased from Gibco (Grand Island, NY, USA). Phenol red-free DMEM, fetal calf serum (FCS), and penicillin–streptomycin mixture were purchased from Invitrogen Carlsbad (Carlsbad, CA, USA). Phosphate buffer saline (PBS) and 3-(4,5-dimethylthiazole-2-yl)-2,5 diphenyltetrazolium bromide (MTT) were purchased from Invitrogen (Waltham, MA, USA). Acetic acid (CH3COOH), potassium dihydrogen orthophosphate (KH_2_PO_4_), sodium hydrogen orthophosphate anhydrous (Na_2_HPO_4_), sodium chloride (NaCl), and sodium hydroxide (NaOH) were purchased from Ajax Finechem (Ajax Finechem Pty Ltd., New South Wales, AUS). Normal human dermal fibroblasts (NHDF), B16 melanoma, and 3T3-L1 adipocyte cell lines were purchased from Lonza Bioscience (New Hampshire, MA, USA). The instruments used were an inverted microscope (Axio Vert. A1 FL LED, ZEISS^®^, Jena, Germany), a 96-well microplate reader (VarioskanTM Flash Multimode Reader, Thermo Scientific^®^, MA, USA), a UV spectrophotometer (UV-1700, Shimazu, Japan), a centrifugation machine (Kubota, Tokyo, Japan), a high-performance liquid chromatography (HPLC) system (Thermo Scientific, Massachusetts, USA), a hot-air oven (France Etuves, France), Whatman paper No.1 (N-1000, Tokyo Rikakikai Co. Ltd., Japan), a rotary evaporator (Buchi, Flawil, Switzerland), a lyophilizer (Scanvac, Lynge, Denmark), and a Sircol^®^ Kit (Biocolor Ltd., Carrickfergus, UK).

### 4.2. Preparation of Cassava Leaf Extract

The 5-month-old cassava leaves were collected from Khon Kaen province in Thailand and dried at 70 °C for 12 h. The dried leaves were blended with 95% ethanol in a 1:4 *w*/*v* ratio using a blender and followed with 24 h maceration. Next, the supernatant was collected using 10 min centrifugation at 10,000 rpm speed and followed with filtration through a filter membrane (Whatman^®^ paper No.1). Subsequently, it was concentrated using a rotary evaporator (at 70 °C temperature and 150 rpm speed) and a lyophilizer (at −80 °C, −30 psi for 24 h). Subsequently, lyophilized powder of the extract was kept at −20 °C until use.

### 4.3. Analysis of Chemical Composition by High-Performance Liquid Chromatography

The chemical constituents in the cassava extract were analyzed using an HPLC system, which comprised an RP-C18 column (Thermo Scientific Hypersil GOLD, particle size 5 µm 4 × 250 mm); a surveyor PDA Plus detector with simultaneous detection at 210, 254, and 280 nm wavelengths; and a gradient mobile phase system of a mixture of deionized (DI) water and methanol within a 60 min run at a flow rate of 0.5 mL/min: the gradient mobile phase program comprised 0% to 95% methanol within 30 min, 95% to 5% methanol with 10 min and hold for 10 min, and 5% to 0% within 5 min and hold for 5 min. A 5 µL injection of methanolic sample solution (10 mg/mL concentration) was performed to obtain the HPLC chromatogram, containing peaks of chemical constituents with the corresponding peak area, retention time, and specific UV spectrum of each. Compared to the reference standard compounds, such as gallic acid, rutin, kaempferol, and apigenin, chemical constituents were identified based on the peak retention time and UV spectrum similarity. A methanolic standard compound mixture stock solution was prepared and diluted to various concentrations (0–50 µg/mL). Subsequently, the quantitation of each chemical constituent was extrapolated from the corresponding standard curves and prepared from standard compound mixture solutions at concentrations of 3.125 to 50 µg/mL after HPLC analysis. The standard curve of each standard was established to obtain a linear equation (y = ax) and linear regression (R^2^) of the relationship, plotted between concentrations versus corresponding peak areas detected at a specific wavelength: 254 nm for rutin, apigenin, and kaempferol; 280 nm for gallic acid.

### 4.4. Determination of UV Screening Capacity

Ethanolic cassava leaf extract solution was prepared at various concentrations (62.5–1000 µg/mL), while the BEMT solution, a reference UV filter, was constituted at different concentrations (6.25–50 µg/mL) in hexane. The absorbance at a wavelength range of 200–400 nm of each sample solution was determined using a double-beam UV spectrophotometer. The UV spectra of each sample, plotted between absorbencies and wavelengths, were obtained to identify the values of γmax—wavelength with maximal absorbance. Moreover, $A_{1-cm}^{1\%}$—absorptivity at γmax on the basis of 1% (*w*/*v*) concentration and 1 cm light path width—was delivered from linear regression curves between absorbances at a λmax wavelength and corresponding concentrations.

### 4.5. Anti-Oxidation by DPPH Radical Scavenging Assay

Using a modified version of the method described by Chaiittianan et al. [10], the methanolic solution of a standard ascorbic acid (vitamin C) and the cassava leaf extract were freshly prepared at designated concentrations to achieve final concentrations in a range of 0–50 µg/mL of standard ascorbic acid and 0–500 µg/mL of cassava leaf extract during the DPPH assay testing. The reaction mixture was constituted in a 96-well plate, comprising 100 µL of test solution (or methanol as the control) mixed with 50 µL of 0.2 mM DPPH (2, 2 diphenyl-1-picryl-hydrazyl) solution, followed by 15 min incubation under light-protected ambient conditions. Next, absorbance was measured at 517 nm wavelength using a microplate reader. The calculation for % inhibition was performed along with the below equation, which established the linear relationship between concentrations and the corresponding percentage of inhibition (% inhibitions). The 50% inhibitory concentration (IC_50_) was extrapolated for the linear relationship at a 50% inhibitory effect.
%inhibition = [(A_0_ − A_sample_)/A_0_] × 100 (1)
where A_0_ is the absorbance of the control, and A_sample_ is the absorbance of the tested sample.

### 4.6. Anti-Melanogenesis in B16 Melanoma Cell Line

Using a modified version of the method described by Phetpornpaisan et al. [48], the B16 at sub-passage#5 was cultured in 10% FBS-supplemented DMEM culture media until 90% confluence before being seeded at a density of 20,000 cells/well into each well of a 96-well plate. After 24 h incubation at maintenance conditions—37 °C temperature with 5% CO_2_ atmosphere—the cells were used.

For the cell viability test via MTT assay, the cells were treated with cassava leaf extract solution of various concentrations and 48 h incubation under maintenance conditions. Next, the supernatant was replaced with 50 µL of MTT solution (0.5 mg/mL) and incubated for 4 h under maintenance conditions until the purple formazan crystals intracellularly formed. Next, the supernatant was gently removed before adding 50 µL dimethyl sulfoxide (DMSO), followed by 1 min mixing. Subsequently, the absorbance (Abs) was measured at 570 nm wavelength using a microplate reader. Compared to the control, the untreated group, the cell viability was calculated and expressed as the % viability of the control using the following equation: % viability = (Abs of sample/Abs of the control) × 100. Thus, the 80–100% cell viability of the control accounted for the non-cytotoxic concentration.

In anti-melanogenesis determination, the cells were treated with various cassava leaf extract concentrations, prepared in 10% (*v*/*v*) FBS and 50 µM forskolin-supplemented DMEM culture medium followed by 48 h incubation under maintenance conditions. The forskolin-treated group was used as the (positive) control and cultured in 10% FBS-supplemented DMEM to extract the melanin pigment; in total, 100 µL NaOH was added into each well, followed by 1 h incubation at 70 °C using hot air and 2 min sonication. After drying the plate, the absorbance was measured at a 405 nm wavelength using a microplate reader. Compared to the control, the results were expressed as the % melanin content.
%Melanin content = (A_sample_/A_0_) × 100 (2)
where A_0_ is the absorbance of the control and A_sample_ is the absorbance of the tested sample.

### 4.7. Collagen Synthesis Enhancement in NHDF Cell Line

Using a modified version of the method described by Phetpornpaisa et al. [48], the collagen synthesis enhancement of the sample was investigated in normal human dermal fibroblast (NHDF). At a density of 20,000 cells/well on a 96-well plate, the NHDF was cultured in 10% FBS-supplemented DMEM culture media and incubated in maintenance conditions (5% CO_2_ at 37 °C) for 24 h before being used in the experiment. A colorimetric measurement was performed using a Sircol^®^ Kit (Biocolor Ltd., Carrickfergus, UK) to quantify the collagen content, comprising dye reagent, acid salt wash reagent, and alkaline reagent.

First, cell viability was investigated using MTT assay; the NHDF cells were treated with sample solutions at various concentrations, freshly prepared in 10% FBS-supplemented DMEM culture medium, and incubated under maintenance conditions for 48 h. Next, the supernatant was replaced by a 50 µL MTT solution (concentration of 5 mg/mL in phosphate buffer saline), followed by another 4 h incubation in the maintenance condition. After time-course completion, the supernatant (MTT solution) was replaced with 50 µL DMSO to dissolve the formazan crystals. Next, the absorbance was measured at a 570 nm wavelength using a microplate reader. Compared to the control (untreated group), the cell viability expressed as % viability was calculated using the following equation:%viability = (A_sample_ /A_0_) × 100 (3)
where A_0_ is the absorbance of the control, and A_sample_ is the absorbance of the tested sample.

Next, the effect on collagen synthesis was determined using vitamin C and rutin as reference compounds; cells were treated with the assigned concentrations of sample solutions, prepared in 10% FBS-supplemented DMEM culture medium, and incubated at the maintenance condition for 48 h. Next, a 20 µL supernatant of each well was sampled and mixed thoroughly with 40 µL dye reagent, then incubated for 10 min at room temperature to form collagen-dyed pellets. The pellets were subsequently collected by 10 min centrifugation at 12,000 rpm speed, and they were rinsed off using a 500 µL acid salt wash reagent. The rinsed pellets were re-collected after another centrifugation and dissolved in a 500 µL alkali reagent. Next, the absorption (Abs) of the solution at a 550 nm wavelength was measured using a microplate reader and used to calculate the % collagen content of the control, the untreated group.
%collagen content = [(A_sample_ − A_blk_)/(A_0_ − A_blk_)] × 100 (4)
where A_0_ is the absorbance of the control, A_sample_ is the absorbance of the tested sample, and A_blk_ is the absorbance of the blank.

### 4.8. Determination of Anti-Adipogenesis in 3T3-L1 Adipocytes

#### 4.8.1. Preadipocyte Maintenance 

Using a modified version of the method described by Chaiittianan et al. [10], the eighth sub-passaged 3T3-L1 murine preadipocytes were cultured in a preadipocyte medium (PM) comprising 10% fetal calf serum (FCS) in Dulbecco’s modified eagle medium (DMEM) containing 1% penicillin-streptomycin, and incubated at the maintenance condition (at 37 °C in a humidified atmosphere of 5% CO_2_) until it reached 90% confluence. They were then used for further experiments. 

#### 4.8.2. Effect on Preadipocyte Viability Using MTT Assay

Preadipocytes were seeded into 96-well plates at a density of 4000 cells/well and cultured in the treatment medium, with various concentrations of BM solution dissolved in the PM culture medium. Next, they were incubated under maintenance conditions for 48 h. Furthermore, the supernatant of each well was replaced by 50 µL of MTT solution (5 mg/mL in phosphate buffer saline) and incubated under maintenance conditions for 4 h. After removing the supernatant (MTT solution), 100 µL of DMSO was added, followed by a 15 min incubation at room temperature. Next, the absorption at 560 nm wavelength was determined using a microplate reader.

#### 4.8.3. Inhibition of Preadipocyte Differentiation and Lipid Accumulation by Using Oil Red O Staining

Cells were seeded into 24-well plates at a density of 24,000 cells/well and cultured until 80% confluence. Next, the differentiation of preadipocytes was induced by 48 h treatment of various media with or without the presence of the sample. The order of media treatment was the differentiation medium I (DMEM, 10% FCS, 1 µM dexamethasone; DEX, 500 µM 3-Isobutyl-1-methylxanthine; IBMX, and 10 µg/mL insulin), differentiation medium II (DMEM, 10% FCS and 10 µg/mL insulin), and differentiation medium III (DMEM, 10% fetal bovine serum; FBS). Subsequently, differentiated adipocytes were obtained and used to investigate the anti-adipogenesis effect of the sample. First, the preadipocytes were treated with differentiated media (type I–III as aforementioned) containing a sample at designated concentrations; meanwhile, the control was treated with a sample-free differentiated medium. Next, the preadipocyte differentiation and intracellular lipid accumulation were determined using the Oil Red O staining technique; the differentiated adipocytes were fixed with 10% (*v*/*v*) formalin for 30 min, rinsed with PBS (pH 7.4), and stained with a freshly prepared 0.5% (*w*/*v*) Oil Red O solution at 37 °C for 1 h. The excess Oil Red O solution was removed and rinsed with 10% isopropanol in phosphate buffer saline. Subsequently, the stained cells were photographed using an inverted microscope. The accumulative lipid content in adipocytes was determined by extracting the retained dye with 100% isopropanol, and quantification was performed by measuring the absorption at 510 nm wavelength using a microplate reader. The inhibitory effect of each sample was expressed as the relative lipid accumulation content compared to the control (% of the control).

### 4.9. Statistical Analysis

All the experiments were performed in triplicate. The results were expressed as the mean ± standard deviation. Statistical analysis using the SPSS version 19 statistical software program by one-way ANOVA analysis of variance and LSD multiple comparisons was conducted. Significance was considered as a *p*-value of less than 0.05 (*p*-value < 0.05).

## 5. Conclusions

The cosmeceutical potential of cassava leaves was evidenced. Interestingly, a rational chemical composition with a high content of rutin, apigenin, and kaempferol, and multiple beneficial bioactivities, including anti-melanogenesis, the enhancement of collagen synthesis, anti-adipogenesis, anti-oxidation, and UV filter capability, were demonstrated. The significant activities of cassava leaves regarding the inhibition of adipogenesis and the enhancement of dermal fibroblast proliferation and collagen synthesis were revealed in association with their chemical composition. Furthermore, synergistic effects among the composited constituents in the cassava leaf were suggested; both BM and rutin had potent enhancing effects on collagen synthesis, similarly to the cassava leaves. Conclusively, cassava leaves are potential natural cosmeceutical active agents due to their chemical composition and multiple bioactivities. However, further investigation and development are encouraged to provide sufficient supporting information for rational and sustainable applications.

## Figures and Tables

**Figure 1 molecules-27-01556-f001:**
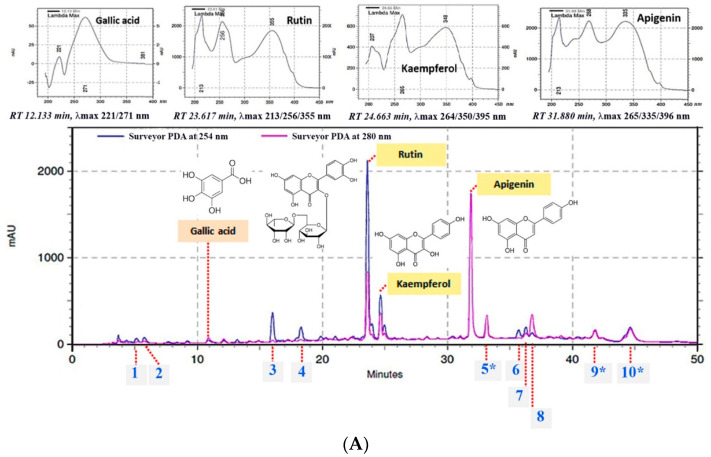
HPLC chromatogram of the ethanolic cassava leaf extract (BM) from HPLC–UV analysis at 254 and 280 nm wavelength detection showed more than 15 peaks of composited compounds. Based on their specific retention times and UV spectra that were identical to that of standard reference compounds, some of them were identified as gallic acid, rutin, kaempferol, and apigenin (**A**). In addition, there were unknown peaks (Unknown 1–10) that showed different UV spectroscopic profiles (**B**). Amongst, some peaks showed typical spectroscopic characters (*).

**Figure 2 molecules-27-01556-f002:**
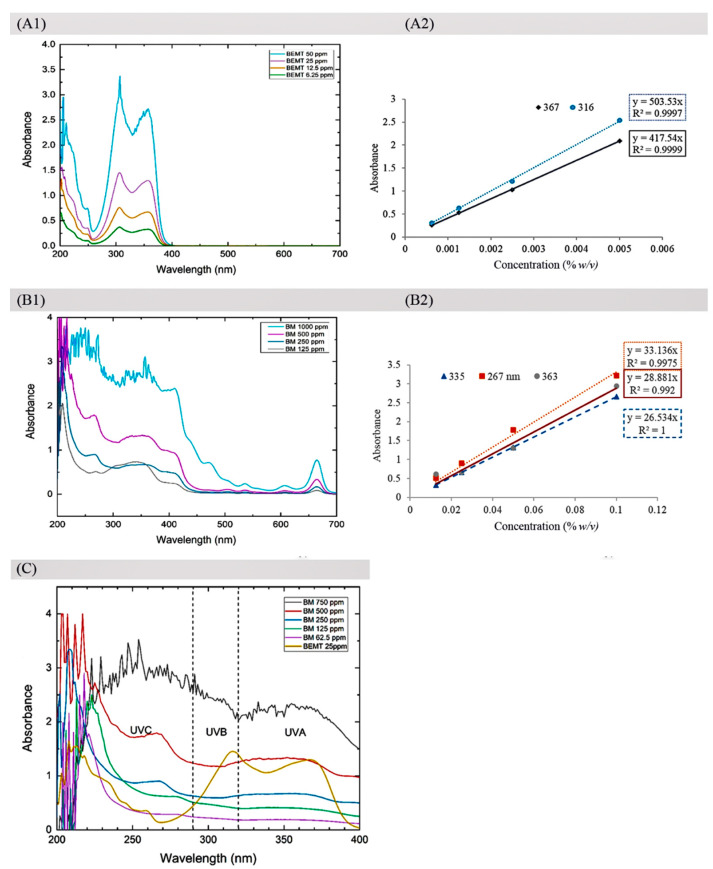
UV absorption spectra of the ethanolic cassava leaf extract (BM) and reference UV filter (BEMT) at various concentrations: broad-spectrum UV absorption of BEMT-covered UV region (200–400 nm) with maximal absorption peaks (λmax) at 316 and 367 nm as the primary and secondary band, respectively (**A1**); absorptivity ($A_{1-cm}^{1\%}$) = 503.53 at 316 nm and 417.54 at 367 nm was manifested (**A2**). By contrast, absorption spectra of BM-covered UV and visible regions (200–700 nm) with λmax at 269, 363, and 355 nm as the primary, secondary, and tertiary bands, respectively (**B1**); $A_{1-cm}^{1\%}$ = 33.136 at 269 nm,28.881 at 363 nm, and 27.534 at 335 nm are displayed (**B2**). In addition, the overlay UV absorption spectra showed the equivalent UV absorption between BM at a concentration of 500 µg/mL and BEMT UV filter at 25 µg/mL (**C**).

**Figure 3 molecules-27-01556-f003:**
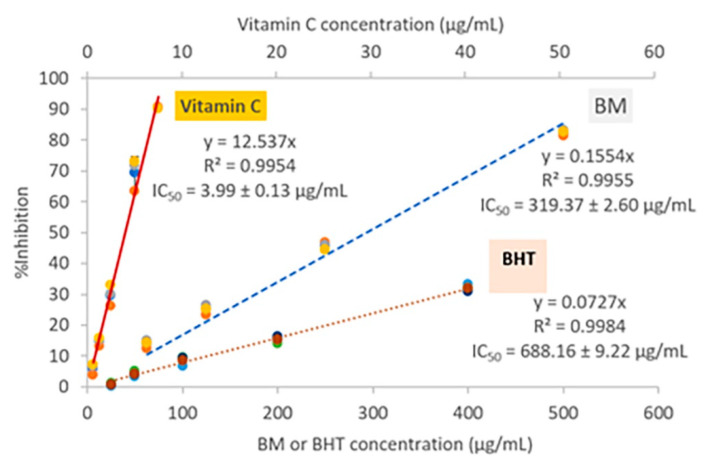
Moderate antioxidative activity of the ethanolic cassava leaf extract (BM) was expressed by a 50% inhibition concentration value (IC_50_) of 319.37 ± 2.60 µg/mL when compared to reference antioxidants, ascorbic acid (vitamin C, IC_50_ 3.99 ± 0.13 µg/mL), and butylated hydroxytoluene (BHT, IC_50_ 688.16 ± 9.22 µg/mL).

**Figure 4 molecules-27-01556-f004:**
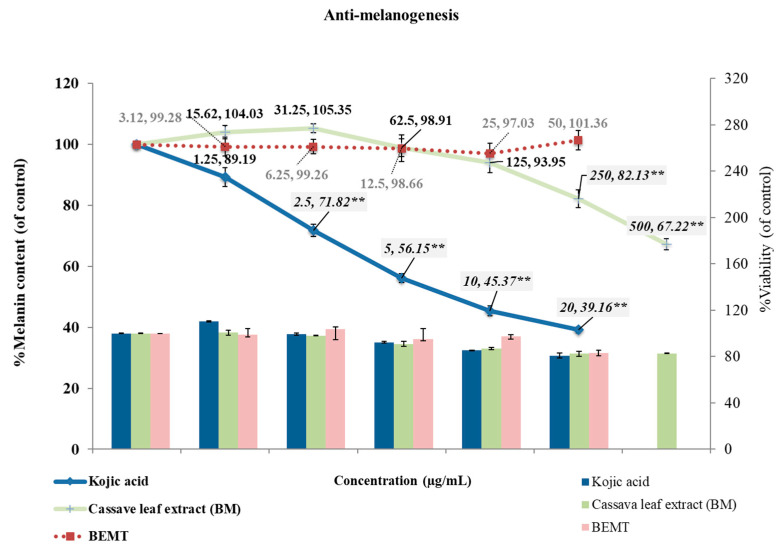
The anti-melanogenesis effect of Kojic acid and the ethanolic cassava leaf extract (BM) was expressed as a melanin content reduction (line) at non-cytotoxic concentrations, with >80% cell viability of the control (bar). A significant effect was detected at 250 and 500 μg/mL BM concentration and 2.5–20 μg/mL Kojic acid concentration, compared to the control (** *p*-value < 0.01). By contrast, BEMT showed neither cytotoxicity nor activity in the 3.12–50 μg/mL concentration range.

**Figure 5 molecules-27-01556-f005:**
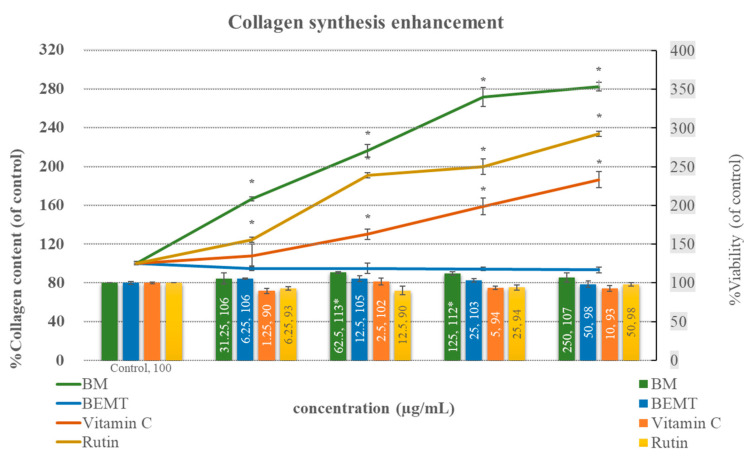
Non-cytotoxicity of all samples was displayed with >80% cell viability of control (bar). Among the samples, significant collagen synthesis enhancement (line) was found in the ethanolic cassava leaf extract (BM), rutin, and vitamin C (* *p*-value < 0.05), in which BM was revealed as the most potent agent, and highly increased the collagen contents compared to rutin and vitamin C. By contrast, BEMT showed no effect on collagen content.

**Figure 6 molecules-27-01556-f006:**
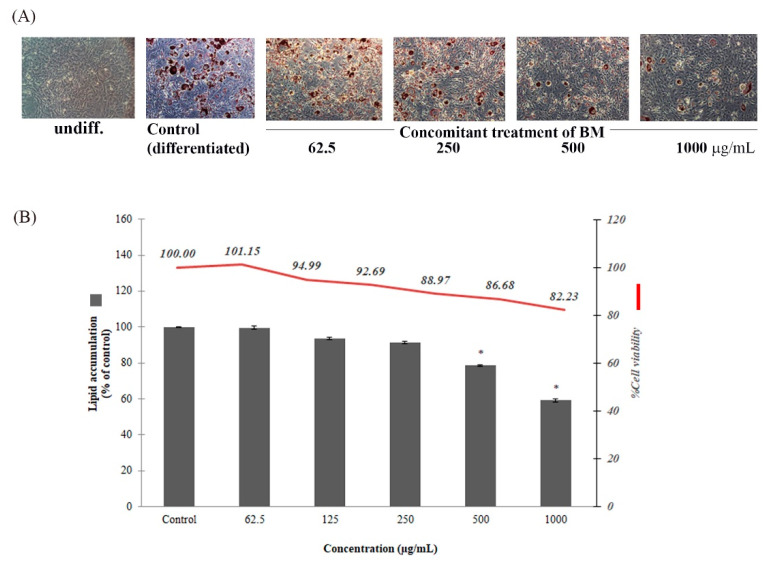
Anti-adipogenesis of the ethanolic cassava leaf extract (BM) was manifested in a dose-dependent manner. Inhibitory effects on preadipocyte differentiation and adipogenesis were expressed as a reduction in Oil Red O-stained cell numbers visualized under an inverted microscope (magnification 20×) (**A**), as well as a reduction in lipid accumulation with statistical significance at concentrations of 500 and 1000 μg/mL when compared to the control (* *p*-value < 0.05) (**B**).

**Table 1 molecules-27-01556-t001:** The content of each constituent extrapolated from a linear regression standard curve (with linear equation y = ax; regression coefficient R^2^) and expressed in a unit of a milligram per gram extract (mg/g extract).

Chemical Constituent	Equation (y = ax)	Linear Regression (R^2^)	Content (mg/g Extract)
Gallic acid	y = 418656x	0.9983	0.31 ± 0.00
Rutin	y = 67947x	0.9974	39.96 ± 0.81
Kaempferol	y = 59296x	0.9962	15.73 ± 0.19
Apigenin	y = 118753x	0.9981	27.70 ± 0.40

## Data Availability

Not applicable.

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
