# Peer review of "Multiple Bioactivities of Manihot esculenta Leaves: UV Filter, Anti-Oxidation, Anti-Melanogenesis, Collagen Synthesis Enhancement, and Anti-Adipogenesis"

_molecules, 2022, doi:10.3390/molecules27051556_

Round 1
Reviewer 1 Report
The manuscript entitled "Multiple bioactivities of Manihot esculenta leaves: UV filter, anti-oxidation, anti-melanogenesis, collagen synthesis enhancement, and anti-adipogenesis", from the authors Manuschanok Jampa, Khaetthareeya Sutthanut, Natthida Weerapreeyakul, Wipawee Tukummee, Jintanaporn Wattanathorn, Suparporn Muchimapura.
The topic of this paper is the useful use of Manihot esculenta leaves, which is a waste material. The results show that this idea can be acceptable and that further research should be done in that direction. That is why I consider that this manuscript should be published in journal Molecules.
Check the Figure 1 B. For peaks 3 and 4, the retention time is the same 16.017 min but different UV spectra. Did this two components came out in one pick? As the instrument recorded different UV spectra, do the UV spectra refer to the ascending and descending part of the peak? Maybe there is an error in marking the RT values for peaks 3 and 4?
Author Response
Dear Reviewer,
I would like to express our sincere thanks for all of the valuable comments and suggestions from you and your kind considerations. Please see the response and revision of manuscript in the attachment.
Best regards,
K. Sutthanut

Reviewer 2 Report
The authors have improved the paper taking into consideration the previous suggestions and comments.
Minor comments:
Line 19 - The solvent used to prepare the extract must be described (cassava leaves ethanolic extract)
Line 20 - as described in the materials and methods sections, authentic standards were used to identify and quantify rutin, apigenin, gallic acid, and kaempferol, by comparison with retention time and UV-spectroscopic characteristics. Therefore, this was not a tentative identification. compounds were identified and quantified by comparing with authentic standards. Please, correct accordingly all over the text (see also lines 87-91).
Lines 24-27 - These two sentences must be improved. Don't use words such as "strong", "significantly", "superior", "fascinating" in excess.
Line 84 and 203 - include "cassava leaves ethanolic extract (BM)"
Line 218 - delete nutrients
lines 241 - 243 - this sentence isn´t clear. Please correct.
Author Response

(The authors gave the same response as above.)

Reviewer 3 Report
The authors present multiple bioactivities of Manihot esculenta leaves: UV filter, anti-oxidation, anti-melanogenesis, collagen synthesis enhancement, and anti-adipogenesis. The manuscript can be accepted in present form.
Author Response
Dear Reviewer,
I would like to express our sincere thanks for all of the valuable comments and suggestions from you and your kind considerations.
Best regards,
K. Sutthanut
This manuscript is a resubmission of an earlier submission. The following is a list of the peer review reports and author responses from that submission.
Round 1
Reviewer 1 Report
This paper describes the biological evaluation of the ethanolic extract of Manihot esculenta (cassava) leaves, namely the antioxidant, UV-screening capacity, anti-melanogenesis, collagen synthesis, and anti-adipogenesis properties. HPLC-DAD assays were also conducted in order to partially characterize the extract.
This paper represents a considerable amount of work; nevertheless, I recommend its rejection because it is badly written, the presentation of the data is hard to understand, it lacks clarity and proper discussion of the results.
The manuscript should be extensively revised and submitted elsewhere.
Importantly, a careful revision of the English language is mandatory.
Some Major considerations
Title
Despite having performed some in vitro assays regarding anti-melanogenesis and anti-adipogenesis, classifying the extract as having multi-mode cosmeceutical properties is quite exaggerated. A revised title is recommended.
Abstract
The abstract is not clear or concise enough; it should be re-written focusing on the scope of the study, the experimental approach, significant results obtained, and major conclusions.
Introduction
The introduction doesn´t provide enough background on the subject.
Authors should clearly refer what is the reason for this research, its significance, and its contribution to new knowledge.
What is the correct name of the species? Cassava is the common name and the plant is also known as manioc, mandioc, or yucca. The correct name is Manihot esculenta (Euphorbiaceae family).
Results
This section should be carefully re-written in order to improve clarity. Some paragraphs are badly written, not understandable, and are confusing.
In particular:
HPLC-DAD was performed to characterize the extract and find out the main constituents that could possibly be correlated with the claimed biological activities. However, only six compounds were identified, namely, gallic acid, rutin, quercitrin (not quercetin – please correct), apigenin, beta-sitosterol, and stigmasterol, together with chlorophyll and carotenoids. The compounds were identified by comparison of their retention time (Rt) and UV data with authentic standards.
- What was the apigenin derivative that you’ve found? Saying that is an apigenin derivative is not scientifically correct.
- Beta-sitosterol has only one double bond, therefore, in the absence of another type of chromophores, it is not correct to say that the maximum wavelength of absorption is 255, 269, and 367 nm. Moreover, something is very wrong with beta-sitosterol UV spectrum. The same consideration is also applied to stigmasterol. These two compounds have a minimum or no UV absorption in wavelengths higher than 230-240 nm. Maybe the peaks in the chromatograms are co-eluted substances with the same Rt. Therefore, if the identification and quantification are not correct it will compromise the presentation and discussion of some results.
- What about the four compounds that you haven´t identified? You should not claim that they could probably be allyl thiocyanate derivatives (discussion line 235) based only on the UV wavelength absorbances without further spectroscopic analysis.
- Do you have a reference standard for chlorophyll?
Consider joining the results and the discussion sections.
Please, think about your results and analyze them properly. After that, you should write one clear results and discussion section.
Author Response
"Please see the attachment"
Thank you
General summary: The overall manuscript was carefully reviewed and improved accordingly throughout the context, such as the consistency citation format, English writing, and clarity of context in all sections, Figures, and Table, addition of more related information in Introduction and Discussion section etc. On behalf of the authors, I would like to express our sincere thanks for all the valuable comments and suggestions which have made an improvement of our manuscript.
A) Importantly, a careful revision of the English language is mandatory.
Response: This was performed accordingly. The manuscript was sent to English professor for editing and improving, and double checked by Grammarly®.
B) Some Major considerations
1) Title
Despite having performed some in vitro assays regarding anti-melanogenesis and anti-adipogenesis, classifying the extract as having multi-mode cosmeceutical properties is quite exaggerated. A revised title is recommended.
Response: The title was changed in accordance with the suggestion.
2) Abstract
The abstract is not clear or concise enough; it should be re-written focusing on the scope of the study, the experimental approach, significant results obtained, and major conclusions.
Response: This had been reviewed and improved accordingly.
3) Introduction
3.1) The introduction doesn´t provide enough background on the subject.
Response: This had been reviewed and improved accordingly by addition of more involving information and background.
3.2) Authors should clearly refer what is the reason for this research, its significance, and its contribution to new knowledge.
Response: This had been reviewed and improved accordingly, which can be seen in the last paragraph of the “introduction” section.
3.3) What is the correct name of the species? Cassava is the common name and the plant is also known as manioc, mandioc, or yucca. The correct name is Manihot esculenta (Euphorbiaceae family).
Response: This had been reviewed and corrected accordingly throughout the manuscript.
4) Results
4.1) This section should be carefully re-written in order to improve clarity. Some paragraphs are badly written, not understandable, and are confusing.
Response: This had been carefully reviewed and improved.
In particular:
HPLC-DAD was performed to characterize the extract and find out the main constituents that could possibly be correlated with the claimed biological activities. However, only six compounds were identified, namely, gallic acid, rutin, quercitrin (not quercetin – please correct), apigenin, beta-sitosterol, and stigmasterol, together with chlorophyll and carotenoids. The compounds were identified by comparison of their retention time (Rt) and UV data with authentic standards.
4.2) What was the apigenin derivative that you’ve found? Saying that is an apigenin derivative is not scientifically correct.
Response: This had been reviewed and improved accordingly throughout the manuscript: revision of several sections including Result, Figure1, Table1, and Discussion was intensively done. This point was discussed and placed in the section “Discussion.”
4.3) Beta-sitosterol has only one double bond, therefore, in the absence of another type of chromophores, it is not correct to say that the maximum wavelength of absorption is 255, 269, and 367 nm. Moreover, something is very wrong with beta-sitosterol UV spectrum. The same consideration is also applied to stigmasterol. These two compounds have a minimum or no UV absorption in wavelengths higher than 230-240 nm. Maybe the peaks in the chromatograms are co-eluted substances with the same Rt. Therefore, if the identification and quantification are not correct it will compromise the presentation and discussion of some results.
Response: The information and analytical data were carefully reviewed and clarified in the Method section. The statement shows the similar spectroscopic property of Beta-sitosterol and stigmasterol which have UV absorption in a range of 230 – 260 nm wavelength which was detected at 254 nm (Figure 1) and the peak areas at 245 nm were used for quantitative analysis to prevent the confounding from the possible co-eluted compounds, particularly for stigmasterol, as shown in the Figure 1.
4.4) What about the four compounds that you haven´t identified? You should not claim that they could probably be allyl thiocyanate derivatives (discussion line 235) based only on the UV wavelength absorbances without further spectroscopic analysis.
Response: This had been reviewed and corrected accordingly: this was removed from the content.
4.5) Do you have a reference standard for chlorophyll?
Response: This was reviewed and clarified: the revision of several sections was done including removal of chlorophyll and carotenoids from the Reagents and instrument section, clarifying the legend of Figure 1 and Discussion section.
4.6) Consider joining the results and the discussion sections.
Please, think about your results and analyze them properly. After that, you should write one clear results and discussion section.
Response: This have been reviewed and considered. After the revision and improvement according to the comments and suggestion, the discussion section has been improved by analyzing the results altogether and it becomes more informative and clear discussing points. Therefore, we decided to present the results and discussion in separate sections for the season of context clarity and conciseness.
Reviewer 2 Report
The manuscript entitled "Multi-mode cosmeceutical property of cassava leaves: UV filter, antioxidation, anti-melanogenesis, collagen synthesis enhancement, and anti-adipogenesis", from the authors Manuschanok Jampa, Khaetthareeya Sutthanut, Wipawee Tukummee, Jintanaporn Wattanathorn, Natthida Weerapreeyakul and Suparporn Muchimapura has been reviewed. The aim of this work was to link the data about chemical composition of the cassava leaf extract with its biological activities. The authors obtained very interesting results and made some important conclusions. The manuscript is very well written and the obtained results are important, so I recommend publishing this manuscript in the journal Molecules.
Author Response
The overall manuscript was carefully reviewed and improved accordingly throughout the context, such as the consistency citation format, English writing, and clarity of context in all sections, Figures, and Table, addition of more related information in Introduction and Discussion section etc. In addition, all queries raised by Reviewer 1 have been responded and conformed in manuscript revision and editing. On behalf of the authors, I would like to express our sincere thanks for all the valuable comments and suggestions for you which have made an improvement of our manuscript.
Round 2
Reviewer 1 Report
I recommend the rejection of the manuscript. Please see the attached file.

Author Response
Dear respectful Reviewer,
On behalf of the authors, I would like to express our sincere thanks and appreciation for your kind comments and suggestions. The revision has been performed accordingly. As the summary of the major revision and improvement, these include the title to be as “Cosmeceutical property of cassava leaves: UV filter, antioxidation, anti-melanogenesis, collagen synthesis enhancement, and anti-adipogenesis,” the entire manuscript English writing proof and editing, as well as re-analyzing of the phytosterol contents and revision of all related sections, re-writing the sections of Abstract, Introduction, and Methodology.
Herewith, I would like to express our sincere thanks in advance for your kind considerations and suggestions.
Sincerely yours,
Assoc. Prof. Khaetthareeya Sutthanut, PhD
